# A genome-wide association study identifies 41 loci associated with eicosanoid levels

Eugene P. Rhee [1✉], Aditya L. Surapaneni[2,3], Pascal Schlosser [3], Mona Alotaibi[4], Yueh-ning Yang[5], Josef Coresh [3], Mohit Jain[4], Susan Cheng [6,7], Bing Yu [5] & Morgan E. Grams [2,3✉]

Eicosanoids are biologically active derivatives of polyunsaturated fatty acids with broad relevance to health and disease. We report a genome-wide association study in 8406 participants of the Atherosclerosis Risk in Communities Study, identifying 41 loci associated with 92 eicosanoids and related metabolites. These findings highlight loci required for eicosanoid biosynthesis, including *FADS1-3*, *ELOVL2*, and numerous *CYP450* loci. In addition, significant associations implicate a range of non-oxidative lipid metabolic processes in eicosanoid regulation, including at *PKD2L1/SCD* and several loci involved in fatty acyl-CoA metabolism. Further, our findings highlight select clearance mechanisms, for example, through the hepatic transporter encoded by *SLCO1B1*. Finally, we identify eicosanoids associated with aspirin and non-steroidal anti-inflammatory drug use and demonstrate the substantial impact of genetic variants even for medication-associated eicosanoids. These findings shed light on both known and unknown aspects of eicosanoid metabolism and motivate interest in several gene-eicosanoid associations as potential functional participants in human disease.

[1] Nephrology Division and Endocrine Unit, Massachusetts General Hospital, Boston, MA, USA. [2] Division of Precision Medicine, New York University School of Medicine, New York, NY, USA. [3] Department of Epidemiology, Johns Hopkins University Bloomberg School of Public Health, Baltimore, MD, USA. [4] Department of Medicine, University of California San Diego, San Diego, CA, USA. [5] Department of Epidemiology, Human Genetics & Environmental Sciences and Human Genetics Center, School of Public Health, University of Texas Health Science Center at Houston, Houston, TX, USA. [6] National Heart, Lung and Blood Institute's and Boston University's Framingham Heart Study, Framingham, MA, USA. [7] Smidt Heart Institute, Cedars-Sinai Medical Center, Los Angeles, CA, USA. ✉email: eprhee@partners.org; morgan.grams@nyulangone.org

Eicosanoids and related oxylipins, hereafter referred to as eicosanoids, are small polar lipid compounds produced via the extensive oxidation of mostly 18- to 22-carbon poly-unsaturated fatty acids (PUFAs)[1–3]. By signaling through cognate receptors, these molecules play key autocrine, paracrine, and endocrine roles across a range of physiological processes, including inflammation, immune activation, thrombosis, and regulation of vascular tone. There are multiple subfamilies of eicosanoids, such as prostaglandins, thromboxanes, leukotrienes, lipoxins, and resolvins, and the production and action of many of these molecules are either targeted or harnessed by common medications such as aspirin and non-steroidal anti-inflammatory drugs (NSAIDs), as well as less common medications such as leukotriene antagonists and prostacyclin analogs.

A more comprehensive understanding of eicosanoid metabolism has the potential to provide new biological insights and opportunities for therapeutic targeting. To date, numerous genome-wide association studies (GWAS) have assessed the genetic determinants of relatively abundant blood metabolites, e.g., amino acids, sugars, organic acids, lipids, etc., measured by liquid chromatography-mass spectrometry and nuclear magnetic resonance spectroscopy[4–13]. These studies have identified many loci associated with blood metabolite levels, highlighting genes that encode enzymes or transporters directly involved in the given metabolite's synthesis, transport, degradation, or excretion. By contrast, GWAS of blood eicosanoids has been more limited, owing in part to technical challenges attributable to their low abundance, dynamic nature, and extensive isometry in chemical structure.

Recently, a directed non-targeted mass spectrometry approach using high mass-accuracy liquid chromatography-mass spectrometry has been developed that can measure hundreds of distinct eicosanoids in plasma, including many not previously documented in humans[14]. Here, we investigate the genetic determinants of these molecules' blood levels in the Atherosclerosis Risk in Communities (ARIC) Study, using a meta-analysis to consider the strongest signals across self-identified European American (EA) and African American (AA) study participants.

## Results

**Study sample.** A total of 8406 individuals in ARIC (Table 1), including 6496 self-identified EA study participants and 1910 self-identified AA study participants, underwent profiling of 223 plasma eicosanoids and related metabolites, including PUFAs (Supplementary Data 1). In the overall sample, the mean age was 56.9 years; 55.3% were female, 26.6% were on treatment for hypertension, and 14.1% had diabetes. The mean estimated glomerular filtration rate was $98.4 \, \text{mL/min/1.73 m}^2$, and 29.6% and 25.6% were on aspirin or non-steroidal anti-inflammatory (NSAID) medications, respectively.

**GWAS identifies 41 genetic loci associated with plasma eicosanoid levels.** We tested the association between genome-wide single-nucleotide polymorphisms (SNPs) imputed to 8,526,654 genetic markers in EA and 14,854,802 markers in the AA populations and plasma eicosanoid and related metabolite levels using an additive genetic model. Using a genome-wide threshold adjusted for the number of eicosanoids ($P < 2.24 \times 10^{-10}$), we identified 41 loci associated with at least one analyte in the meta-analysis (Table 2, Supplementary Data 2). Several loci had numerous associations, including FADS1-3 ($n = 40$), SLCO1B1 ($n = 25$), PKD2L1 ($n = 11$), and ELOVL2 ($n = 6$). When the EA and AA cohorts were analyzed separately, 9 of 41 loci identified by meta-analysis had significant associations in both the EA and AA cohorts, 25 had significant associations in the EA cohort only, 1 had a significant association in the AA cohort only, and 6 did not have an association in either the EA or AA cohort (Supplementary Data 2). Notably, effect sizes were highly consistent for these loci across ancestry (Supplementary Fig. 1). Of the 223 measured eicosanoids, 92 (41%) had at least one significant genetic association, including 31 with more than one significant genetic association. Manhattan plots for all significant associations are shown in Supplementary Figs. 2–9. Together, these findings implicate a range of loci that encode enzymes and transporters that impact various aspects of eicosanoid metabolism (Fig. 1).

**PUFA biosynthesis.** Eicosanoids are derived from PUFAs—including omega-6 fatty acids such as arachidonic acid (C20:4n-6) and adrenic acid (C22:4n-6), and omega-3 fatty acids such as eicosapentaenoic acid (C20:5n-3), docosapentaenoic acid (C22:5n-3), and docosahexaenoic acid (C22:6n-3)—that are synthesized from the progressive desaturation and elongation of the essential fatty acids linoleic acid (C18:2n-6) and α-linolenic acid (C18:3n-3). PUFA desaturation is catalyzed by the delta-5 and delta-6 desaturases, which are encoded by FADS1-3, whereas PUFA elongation is catalyzed by several elongases, including the very long fatty acid elongase encoded by ELOVL2. In the literature, both FADS1-3 and ELOVL2 have been consistently associated with levels of PUFAs and PUFA-containing lipids such as triacylglycerols and phospholipids, the latter of which serve as the reservoir for PUFAs that are converted into eicosanoids[15, 16]. Several of the significant associations in our analysis recapitulate these published findings; more specifically, we found that SNPs in FADS1-3 are significantly associated with blood levels of the PUFAs arachidonic acid, adrenic acid, and eicosapentaenoic acid and that SNPs in ELOVL2 are associated with adrenic acid, eicosapentaenoic acid, docosapentaenoic acid, and docosahexaenoic acid. In addition, we found that SNPs in these loci are associated with numerous downstream eicosanoids spanning several subfamilies, including prostanoids (prostaglandins and

| Table 1 ARIC study sample. | | | |
|---|---|---|---|
| | **All** | **EA** | **AA** |
| *n* | 8406 | 6496 | 1910 |
| Age, years | 56.9 (5.7) | 57.2 (5.7) | 56.0 (5.8) |
| Female, *n* (%) | 4647 (55.3%) | 3435 (52.9%) | 1212 (63.5%) |
| Hypertension, *n* (%) | 2226 (26.6%) | 1369 (21.1%) | 857 (45.4%) |
| Systolic blood pressure, mmHg | 121.2 (18.8) | 119.7 (17.8) | 126.3 (21.0) |
| Diabetes, *n* (%) | 1180 (14.1%) | 735 (11.3%) | 445 (23.5%) |
| Current smoker, *n* (%) | 1870 (22.3%) | 1387 (21.4%) | 483 (25.4%) |
| Estimated glomerular filtration rate, mL/min/1.73 m$^2$ | 98.4 (16.6) | 98.6 (15.8) | 97.8 (19.1) |
| Aspirin use | 2479 (29.6%) | 2144 (33.0%) | 335 (17.7%) |
| NSAID use | 2144 (25.6%) | 1644 (25.3%) | 500 (26.5%) |
| Data represent means (standard deviation) unless otherwise noted. | | | |

**Table 2 Significant loci associated with blood eicosanoid levels.**

| Locus | Eicosanoid | SNP | rsid | Beta | SE | P-value | EA CAF | AA CAF | Pos |
|---|---|---|---|---|---|---|---|---|---|
| FADS1-3 | FFA_Arachidonic Acid_a | chr11:61800281:C:A | rs174544 | 0.54 | 0.02 | 6.60E-207 | 0.29 | 0.07 | 3'UTR |
| SLCO1B1 | 11t LTD4 | chr12:21178615:T:C | rs4149056 | 0.54 | 0.02 | 1.47E-124 | 0.16 | 0.03 | missense |
| PKD2L1 | 11-hydroxy-9-octadecenoate; 10-hydroxy-11-octadecenoate | chr10:100315722:G:A | rs603424 | -0.41 | 0.02 | 1.94E-113 | 0.18 | 0.68 | intron |
| ACOT4/ACOT6 | Dihydroxydocosapentaenoic acid | chr14:73610482:G:T | rs11511359 | -0.34 | 0.02 | 1.99E-64 | 0.21 | 0.06 | intergene |
| ACSM6 | 12,13-diHOME; 9,10-diHOME | chr10:95215869:A:G | rs612490 | -0.24 | 0.02 | 1.96E-55 | 0.57 | 0.53 | intron |
| CYP4Z2P/CYP4A11 | 17-HETE; 18(+/-)-HETE; 20-HETE | chr1:46917369:A:T | rs4507958 | -0.31 | 0.02 | 3.02E-52 | 0.14 | 0.36 | intergene |
| ACAD11 | tetranor 12(R)-HETE_a | chr3:132611423:G:T | rs111910466 | 0.89 | 0.06 | 5.10E-52 | 0.01 | 0.03 | intron |
| RPL7APF2/CYP2C9 | 13-oxoODE_a | chr10:94898738:G:A | rs7910609 | 0.45 | 0.03 | 1.43E-46 | 0.06 | 0.06 | intergene |
| CYP3A5 | 11-dehydro-2,3-dinor-TXB2_b | chr7:99672916:T:C | rs776746 | 0.35 | 0.02 | 4.27E-46 | 0.93 | 0.30 | intron |
| CYP2C18 | 19(R)-HETE; 20-HETE | chr10:94698005:G:C | rs12773884 | -0.29 | 0.02 | 1.05E-41 | 0.15 | 0.17 | intron |
| CYP1B1/CYP1B2 | 8-iso-PGA1; PGA1_a | chr8:142901337:A:G | rs4736317 | -0.19 | 0.02 | 4.38E-32 | 0.56 | 0.81 | intergene |
| SRD5A2 | 11-dehydro-2,3-dinor-TXB2_a | chr2:31585905:T:A | rs559555 | 0.18 | 0.02 | 4.54E-32 | 0.56 | 0.49 | intron |
| CYP4F2 | 8-iso-PGF1a; 8-iso-PGF1b; PGF1beta_a | chr19:15879621:C:T | rs2108622 | -0.22 | 0.02 | 6.77E-32 | 0.28 | 0.09 | missense |
| SLC27A2 | 11t LTD4 | chr15:50184941:C:T | rs1365505 | 0.19 | 0.02 | 4.53E-31 | 0.67 | 0.41 | intron |
| HSD17B12/ALKBH3 | 12,13-diHOME; 9,10-diHOME | chr11:43856935:G:A | rs57635800 | 0.18 | 0.02 | 1.42E-27 | 0.29 | 0.42 | intergene |
| SCCPDH | 5(S)-HETrE_b | chr1:246727353:C:T | rs35736382 | 0.25 | 0.02 | 2.26E-27 | 0.09 | 0.34 | intron |
| PECR | 12S-HpETE_b | chr2:216039296:A:T | rs9288513 | 0.25 | 0.02 | 6.93E-26 | 0.10 | 0.23 | missense |
| TMEM258 | 17-HETE; 18(+/-)-HETE; 20-HETE | chr11:61790354:T:C | rs102274 | -0.18 | 0.03 | 5.75E-25 | 0.33 | 0.08 | intron |
| EPHX2 | 12,13-EpOME; 9,10-EpOME | chr8:27494221:G:A | rs10091679 | 0.27 | 0.03 | 1.73E-23 | 0.10 | 0.05 | intron |
| ARPC1A/ARPC1B | 13,14-DiHDPA; 16,17-DiHDPA; 19,20-DiHDPA_b | chr7:99367992:G:A | rs143524414 | -0.32 | 0.03 | 2.12E-21 | 0.07 | 0.01 | intergene |
| ELOVL2 | FFA_Eicosapentaenoic Acid_d | chr6:10996933:C:T | rs9295741 | 0.15 | 0.02 | 3.28E-20 | 0.42 | 0.21 | intron |
| POR | 15 oxoEDE | chr7:75917574:G:A | rs3898649 | 0.16 | 0.02 | 5.33E-18 | 0.26 | 0.84 | intron |
| LINC02732 | 8-iso-PGA1; PGA1_a | chr11:110355722:T:C | rs969680 | -0.14 | 0.02 | 5.08E-17 | 0.25 | 0.44 | intron |
| CYP3A137P/CYP3A43 | 13,14-DiHDPA; 16,17-DiHDPA; 19,20-DiHDPA_a | chr7:99823462:G:A | rs62247956 | -0.35 | 0.04 | 6.52E-17 | 0.05 | 0.01 | intergene |
| LINC01835/CYP4F36P | 9S-HpOTrE | chr19:15868934:C:A | rs77420750 | -0.15 | 0.02 | 9.96E-17 | 0.28 | 0.09 | intergene |
| CYP2C8/LOC107984257 | 5,6-diHETrE_b | chr10:95083926:C:T | rs528961621 | 0.19 | 0.02 | 1.03E-16 | 0.12 | 0.16 | intergene |
| SULT2A1 | 13,14-DiHDPA; 16,17-DiHDPA; 19,20-DiHDPA_b | chr19:47886106:G:A | rs296361 | -0.19 | 0.02 | 5.73E-16 | 0.15 | 0.02 | intron |
| UGT2B15 | 15-epi-PGA1; PGA1 | chr4:68670366:A:C | rs1902023 | -0.13 | 0.02 | 2.09E-15 | 0.47 | 0.58 | missense |
| THEM4/KRT8P28 | tetranor 12(R)-HETE_b | chr1:151944639:G:A | rs9943251 | -0.13 | 0.02 | 6.76E-15 | 0.31 | 0.31 | intergene |
| FAAH | 13,14-DiHDPA; 16,17-DiHDPA; 19,20-DiHDPA_b | chr1:46405089:C:A | rs324420 | -0.14 | 0.02 | 1.20E-14 | 0.20 | 0.37 | missense |
| ACOT1/ACOT2 | HXA3; HXB3 | chr14:73566890:A:G | rs11626972 | -0.12 | 0.02 | 4.81E-14 | 0.42 | 0.36 | intergene |
| UGT2B7 | Dihydroxydocosapentaenoic acid | chr4:69099784:T:C | rs4293848 | -0.12 | 0.02 | 6.19E-14 | 0.47 | 0.71 | intron |
| ABCG8 | 8-iso-PGF1a; 8-iso-PGF1b; PGF1beta_a | chr2:43847292:C:T | rs4245791 | 0.13 | 0.02 | 5.26E-13 | 0.69 | 0.86 | intron |
| PNPLA3 | FFA_Adrenic Acid_a | chr22:43928850:C:T | rs738408 | 0.14 | 0.02 | 2.29E-12 | 0.23 | 0.14 | synonymous |
| ABCC3 | 11t LTD4 | chr17:50676879:T:G | rs12943812 | -0.12 | 0.02 | 4.38E-12 | 0.36 | 0.11 | intron |
| SLC22A10 | MCTR2 | chr11:63295020:G:A | rs59922153 | -0.24 | 0.04 | 8.29E-12 | 0.06 | 0.02 | intron |
| ABCC1 | 13-HODE; 9-HODE_b | chr16:16013420:G:A | rs4781721 | 0.11 | 0.02 | 8.40E-12 | 0.62 | 0.23 | intron |
| TTC27 | 11-dehydro-2,3-dinor-TXB2_a | chr2:32657977:G:A | rs12614750 | -0.24 | 0.03 | 9.25E-12 | 0.06 | 0.04 | intron |
| HSD17B4 | 13,14-DiHDPA; 16,17-DiHDPA; 19,20-DiHDPA_b | chr5:119476953:T:C | rs3850200 | 0.16 | 0.02 | 1.88E-11 | 0.08 | 0.38 | intron |
| ABCC2 | 9-oxoOTrE_b | chr10:99835096:C:T | rs55672373 | 0.22 | 0.03 | 1.32E-10 | 0.06 | 0.06 | intron |
| ADH1A/ADH1B | 17-HETE; 18(+/-)-HETE; 20-HETE | chr4:99304835:T:C | rs1693458 | -0.13 | 0.02 | 1.97E-10 | 0.83 | 0.76 | intergene |

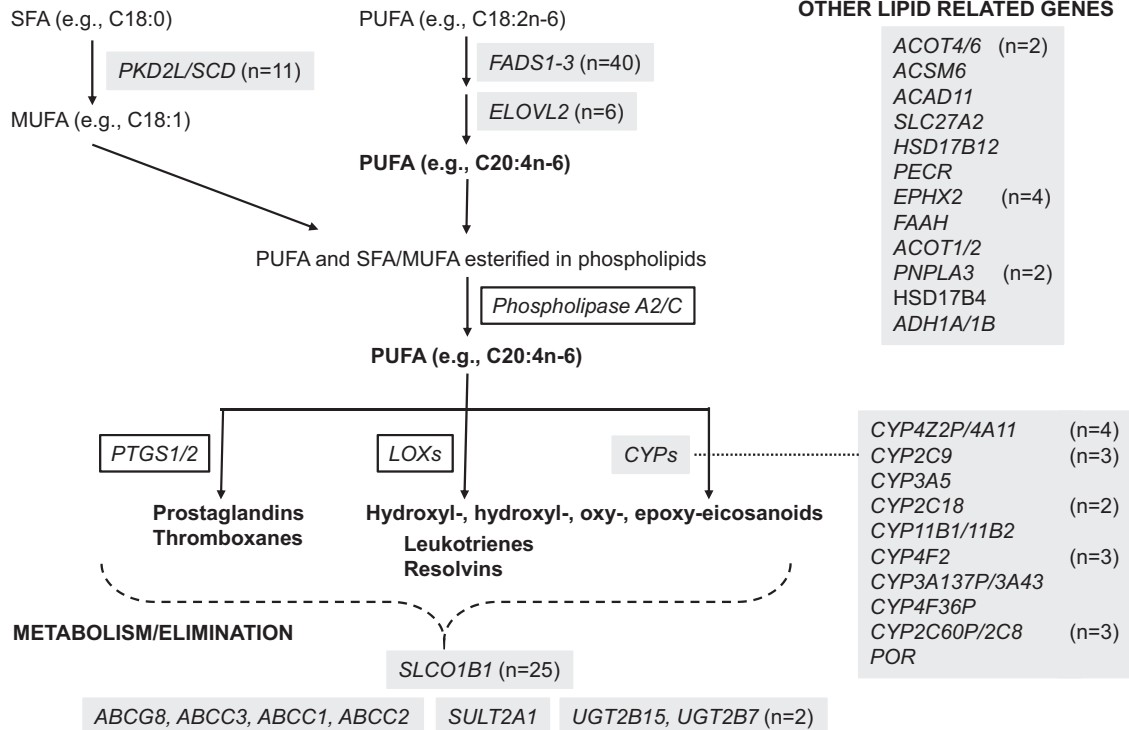

**Fig. 1 Significant GWAS associations and their potential relation to eicosanoid metabolism.** Significant loci are highlighted in gray, with multiple eicosanoid associations at a given locus noted by the number in parentheses. Eicosanoids and related metabolites that were measured by liquid chromatography-mass spectrometry are bolded, whereas metabolites that were not measured are not bolded. SFA saturated fatty acid, MUFA monounsaturated fatty acid, PUFA polyunsaturated fatty acid.

thromboxanes) and hydroperoxyl-, hydroxyl-, oxo-, and epoxy-eicosanoids derived from different PUFAs (Supplementary Data 2).

**Lipid oxidation**. The biosynthesis of eicosanoids requires oxidation of PUFAs, with distinct pathways that include cyclooxygenases, lipoxygenases, and cytochrome P450 enzymes (Fig. 1). We identified 19 significant eicosanoid associations at nine cytochrome P450 loci—CYP4Z2P/CYP4A11 ($n = 4$), CYP2C9 ($n = 3$), CYP3A5, CYP2C18 ($n = 2$), CYP11B1/CYP11B2, CYP4F2 ($n = 3$), CYP3A137P/CYP3A43, CYP4F36P, and CYP2C60P/CYP2C8 ($n = 3$)—as well as at POR, which encodes a cytochrome P450 oxidoreductase that donates electrons directly from NADPH to cytochrome P450 enzymes. In addition, we identified a significant association at ADH1A/ADH1B, which encodes an alcohol dehydrogenase that oxidizes lipid peroxides in addition to alcohols. In contrast to FADS1-3 and ELOVL2, these loci were not significantly associated with any PUFAs. Instead, consistent with their biochemical functions, these loci were predominantly associated with hydroperoxyl-, hydroxyl-, and oxo-eicosanoids derived from different PUFAs, as well as prostaglandins and a hepoxilin (Supplementary Data 2). Significant associations were also observed at EPHX2 ($n = 4$), which encodes epoxide hydrolase 2, an enzyme that converts epoxy lipids to their corresponding diols.

Interestingly, we did not identify any significant associations in PTGS1 and PTGS2, which encode COX1 and COX2, respectively, or in any loci encoding lipoxygenases. As a sensitivity analysis, we examined the association of all SNPs within 500 Mb of PTGS1 and PTGS2 and blood eicosanoids. Even at a relaxed statistical threshold, adjusted only for the number of eicosanoids and the ~6000 SNPs in these regions ($P < 0.05/223/6000 = 3.7 \times 10^{-8}$), no significant associations were observed (Supplementary Data 3).

**Other facets of lipid metabolism**. Several of the significant loci identified by GWAS encode enzymes involved in non-oxidative steps of lipid metabolism. Although not clearly involved in the known, canonical steps of eicosanoid biosynthesis, we highlight them herein because of the biological plausibility of interactions across the lipidome. These loci include several genes involved in fatty acyl-CoA metabolism: ACOT4/ACOT6 and ACOT1/ACOT2, which encode acyl-CoA thioesterases; ACSM6, which encodes a medium chain acyl-CoA synthetase; ACAD11, which encodes an acyl-CoA dehydrogenase; SLC27A2, which encodes a long-chain acyl-CoA ligase; and PECR, which encodes an acyl-CoA reductase. Additional associations were at HSD17B12 and HSD17B4, which encode hydroxysteroid dehydrogenases, FAAH, which encodes a fatty amide hydrolase, and PNPLA3, which encodes a triacylglycerol lipase. Finally, we note that several of the strongest associations were at PKD2L1, which encodes an integral membrane protein of the polycystin family with no obvious lipid-related function. However, a recent study showed that different alleles at the index SNP at PKD2L1, rs603424, are associated with differential chromatin accessibility and gene expression of the downstream gene SCD in adipocytes[17].

**Loci involved in metabolite clearance**. In addition to identifying loci involved in lipid desaturation, elongation, oxidation, and hydrolysis, our findings outline several mechanisms for the transport of eicosanoids across cellular membranes, including the elimination of eicosanoids from blood. For example, SLCO1B1 encodes the organic anion transporting polypeptide 1B1, a liver-specific protein that transports compounds from the blood into the liver so that they can be cleared from the body. Our results implicate this protein in the disposition of a wide range of eicosanoids, including hydroperoxyl-, hydroxyl-, and epoxy-eicosanoids derived from different PUFAs, prostaglandins, leukotrienes, and several PUFAs

(Supplementary Data 2). More widely expressed than *SLCO1B1*, *ABCG8*, *ABCC3, ABCC1*, and *ABCC2* also encode transporters, in this case as members of the superfamily ATP-binding cassette transporters that bind and hydrolyze ATP to enable active transport of a wide range of compounds across cell membranes.

In addition to underlining a role for transcellular transporters in eicosanoid metabolism, our results delineate a role for eicosanoid conjugation that increases their water solubility, thereby facilitating excretion into either urine or bile. For example, *SULT2A1*, which encodes a sulfotransferase, was associated with a docosanoid, whereas *UGT2B15* and *UGT2B7*, which encode glycosyltransferases, were associated with a prostaglandin, a PUFA derivative, and a hydroperoxyl-eicosanoid (Supplementary Data 2).

**Transcriptome-wide association study corroborates genes implicated by eicosanoid GWAS**. Thirty-five of the 41 loci associated with blood eicosanoids had at least one significant transcriptome-wide association study (TWAS) association in a variety of tissues from participants from the Genotype-Tissue Expression Project ("Methods", Supplementary Data 4) at a *P*-value threshold of $4.63 \times 10^{-6}$ (10,806 tests accounting for the number of transcripts within a 500 Kb window of the loci in each tissue). For the large majority of loci, TWAS hits included the genes assigned to each top SNP, as shown in Table 2 (for SNPs that are intergenic, both the closest upstream and downstream genes are assigned). However, in some cases, the top TWAS hits raised the possibility that other genes may underlie the locus-eicosanoid association. In one notable example, the most significant TWAS hit for the locus annotated as *PKD2L1* (index SNP rs603424) was for *SCD* expression in subcutaneous adipose tissue ($P = 1.79 \times 10^{-68}$), thus corroborating the recent literature[17]. Similarly, the most significant TWAS hit for the locus annotated as *ARPC1A/ARPC1B*, which encodes actin-binding proteins, was *CYP3A7* expression in the adrenal gland. We also note that for several of the cytochrome P450-associated loci, TWAS highlighted numerous potential downstream cytochrome P450 genes. For example, the signal in *CYP3A5* (index SNP rs776746) was associated with gene expression at *CYP3A5*, as well as at *CYP3A7*, *CYP3A43*, *CYP3A51P*, and *CYP3A4*. Similarly, *CYP2C18* (index SNP rs12773884) was associated with gene expression at *CYP2C18*, as well as at *CYP2C19*, *CYP2C9*, and *CYP2C8*, and *CYP4F2* (index SNP rs2108622) was associated with gene expression at *CYP4F2*, as well as at *CYP4F11 and CYP4F12*. These findings are not surprising given the known clustering of these gene families but nevertheless indicate that assignment of eicosanoids as substrates or products of specific cytochrome P450 enzymes may not be feasible based on GWAS associations.

**Eicosanoids regulated by multiple genetic loci**. As noted, 31 eicosanoids had more than one significant GWAS association. Of these, 12 had three or more significant GWAS associations (Table 3), underscoring how various aspects of metabolism can impact an eicosanoid's circulating levels. For example, 13-HpODE, 20cooh AA_c, and 5S-HpEPE all had significant associations at *FADS1-3* (PUFA biosynthesis), one or more cytochrome P450 locus (lipid oxidation), and *SLCO1B1* (clearance). The eicosanoid 13,14-DiHDPA; 16,17-DiHDPA; 19,20-DiHDPA_a had significant associations at loci encoding a fatty amide hydrolase (*FAAH*), a hydroxysteroid dehydrogenase (*HSD17B4*), a sulfotransferase (*SULT2A1*), and actin-related proteins (*ARPC1A/ARPC1B*); as noted, TWAS indicates that the latter association may be attributable to a cytochrome P450 enzyme.

**Impact of aspirin and NSAIDs on blood eicosanoids**. To provide perspective on the impact of genetic variants versus environmental factors on blood eicosanoids, we assessed eicosanoid associations with aspirin and NSAID use (Fig. 2). Adjusting for the number of analytes measured ($P < 2.24 \times 10^{-4}$), aspirin use was associated with two eicosanoids, whereas NSAID use was associated with 15 eicosanoids (Table 4). Of the 17 eicosanoids associated with aspirin or NSAID use, 11 had at least one significant GWAS hit.

For all significant GWAS hits, we also reanalyzed the association between each SNP and eicosanoid in models adjusted for aspirin and NSAID use. As shown in Supplementary Fig. 10, adjustment for medication use had no discernible impact on the strength of GWAS associations.

## Discussion

Because of the multiple physiological and biological actions of eicosanoids, an improved understanding of eicosanoid metabolism has the potential to provide insight into both human health and disease. Leveraging the assay and identification of 223 eicosanoids in a large, population-based cohort study, this study identifies dozens of associations between common genetic variants and blood eicosanoid levels. In addition to highlighting loci known to be required for the biosynthesis of eicosanoids, these associations implicate a range of lipid metabolic processes and underscore the important role of clearance mechanisms in the regulation of eicosanoids.

Of the 41 loci highlighted by our analysis, the *FADS1-3* locus had the largest number of associations, consistent with the rate-limiting role that the encoded delta-5 and delta-6-desaturases play in PUFA biosynthesis. The index SNP in *FADS1-3*, rs174544, is a non-coding SNP; the minor allele at this and other SNPs in close linkage disequilibrium are known to be associated with lower expression of the delta-5 and delta-6 desaturases. In our analysis, we find that these alleles associated with lower *FADS1-3* expression are generally associated with lower levels of arachidonic acid-derived eicosanoids and higher levels of eicosanoids derived from fatty acids upstream of arachidonic acid, such as linoleic acid-derived HODEs and dihomo-gamma-linolenic acid-derived HETrEs. Notably, these same *FADS1-3* variants have been associated with a broad range of traits, including asthma[18], rheumatoid arthritis[19], white blood cell count[20], and pulse pressure[21]. Our GWAS findings nominate lower and higher levels of a range of bioactive eicosanoids downstream of PUFAs as potential causal mediators of these clinical phenotypes.

As with *FADS1-3*, associations at *ELOVL2* and several loci encoding redox active enzymes, particularly cytochrome P450 enzymes, also correspond to known eicosanoid biosynthetic pathways. By contrast, several significant associations are at loci encoding enzymes broadly relevant to lipid metabolism but not directly involved in eicosanoid production. Several of the strongest associations are at rs603424, which is within the *PKD2L1* locus, but which has been linked to *SCD* gene expression (including in our TWAS analyses)[17]. *SCD* encodes the delta-9 desaturase, which converts non-essential saturated fatty acids into monounsaturated fatty acids (MUFAs). MUFA synthesis is a vital step in de novo lipogenesis, whereby excess energy is stored as triglyceride and is completely distinct from the delta-5 and delta-6 desaturase-mediated desaturation of essential fatty acids into PUFAs[22]. In the literature, rs603424 has been linked to blood levels of MUFAs and MUFA containing lipids[23], as well as cardiometabolic phenotypes such as low-density lipoprotein cholesterol levels[24], glycated hemoglobin levels[18], blood pressure[25], and coronary artery disease[26]. This locus has not been associated with PUFA levels in humans. However, transgenic expression of the ortholog *SCD1* in mice increases both MUFA and PUFA tissue content, leading the authors to hypothesize that increased MUFA biosynthesis regulates PUFA utilization;[27] conversely, PUFAs are known to inhibit the

**Table 3 Eicosanoids with three or more significant genetic associations.**

| Eicosanoid | Locus | SNP | rsid | Beta | SE | P-value |
|---|---|---|---|---|---|---|
| 11t LTD4 | SLCO1B1 | chr12:21178615:T:C | rs4149056 | 0.54 | 0.02 | 1.47E-124 |
| | ABCC3 | chr17:50676879:T:G | rs12943812 | −0.12 | 0.02 | 4.38E-12 |
| | SLC27A2 | chr15:50184941:C:T | rs1365505 | 0.19 | 0.02 | 4.53E-31 |
| 13,14-DiHDPA; 16,17-DiHDPA; 19,20-DiHDPA_b | CYP3A137P/ CYP3A43 | chr7:99823462:G:A | rs62471956 | −0.35 | 0.04 | 6.52E-17 |
| | SLCO1B1 | chr12:21178615:T:C | rs4149056 | −0.19 | 0.02 | 1.03E-16 |
| | ELOVL2 | chr6:10996933:C:T | rs9295741 | −0.12 | 0.02 | 3.18E-13 |
| 13-HpODE | FADS1-3 | chr11:61820833:A:G | rs174564 | 0.12 | 0.02 | 1.30E-12 |
| | SLCO1B1 | chr12:21178615:T:C | rs4149056 | 0.28 | 0.02 | 2.92E-34 |
| | CYP4B1/CYP4Z2P | chr1:46822094:G:C | rs4660960 | 0.14 | 0.02 | 9.00E-11 |
| | CYP4F2 | chr19:15879621:C:T | rs2108622 | −0.14 | 0.02 | 8.12E-15 |
| 13,14-DiHDPA; 16,17-DiHDPA; 19,20-DiHDPA_a | ARPC1A/ARPC1B | chr7:99367992:G:A | rs143524414 | −0.32 | 0.03 | 2.12E-21 |
| | SULT2A1 | chr19:47886106:G:A | rs296361 | −0.19 | 0.02 | 5.73E-16 |
| | FAAH | chr1:46405089:C:A | rs324420 | −0.14 | 0.02 | 1.20E-14 |
| | HSD17B4 | chr5:119476953:T:C | rs3850200 | 0.16 | 0.02 | 1.88E-11 |
| 17-HETE; 18(+/-)-HETE; 20-HETE | CYP4Z2P/CYP4A11 | chr1:46917369:A:T | rs4507958 | −0.31 | 0.02 | 3.02E-52 |
| | TMEM258 | chr11:61790354:T:C | rs102274 | −0.18 | 0.02 | 5.75E-25 |
| | ADH1A/ADH1B | chr4:99304835:T:C | rs1693458 | −0.13 | 0.02 | 1.97E-10 |
| 20cooh AA_c | FADS1-3 | chr11:61820833:A:G | rs174564 | −0.38 | 0.02 | 1.02E-114 |
| | SLCO1B1 | chr12:21233084:A:G | rs11045885 | 0.16 | 0.02 | 1.64E-16 |
| | CYP4Z2P/CYP4A11 | chr1:46908367:T:C | rs6687264 | −0.30 | 0.02 | 1.46E-42 |
| 5(S)-HETrE_b | SLCO1B1 | chr12:21178615:T:C | rs4149056 | −0.18 | 0.02 | 1.69E-14 |
| | SCCPDH | chr1:246727353:C:T | rs35736382 | 0.25 | 0.02 | 2.26E-27 |
| | PECR | chr2:216060244:C:T | rs9288514 | 0.19 | 0.03 | 8.56E-13 |
| 5S-HpEPE | CYP4F2 | chr19:15879621:C:T | rs2108622 | −0.14 | 0.02 | 7.78E-14 |
| | FADS1-3 | chr11:61822009:A:G | rs28456 | 0.13 | 0.02 | 1.49E-13 |
| | SLCO1B1 | chr12:21178615:T:C | rs4149056 | 0.24 | 0.02 | 3.70E-24 |
| 8-iso-PGA1; PGA1_a | CYP11B1/CYP11B2 | chr8:142901337:A:G | rs4736317 | −0.19 | 0.02 | 4.38E-32 |
| | SRD5A2 | chr2:31585905:T:A | rs559555 | −0.12 | 0.02 | 1.79E-15 |
| | LINC02732 | chr11:110355722:T:C | rs969680 | −0.14 | 0.02 | 5.08E-17 |
| FFA_Adrenic Acid_a | FADS1-3 | chr11:61812288:T:C | rs174555 | −0.14 | 0.02 | 4.34E-15 |
| | SLCO1B1 | chr12:21215863:T:A | rs2900478 | 0.19 | 0.02 | 6.82E-17 |
| | ELOVL2-AS1/ SMIM13 | chr6:11087547:C:T | rs9366722 | 0.13 | 0.02 | 4.51E-11 |
| | PNPLA3 | chr22:43928850:C:T | rs738408 | 0.14 | 0.02 | 2.29E-12 |
| FFA_Eicosapentaenoic Acid_d | FADS1-3 | chr11:61820833:A:G | rs174564 | −0.27 | 0.02 | 1.37E-53 |
| | SLCO1B1 | chr12:21227696:A:T | rs4149083 | 0.16 | 0.02 | 2.03E-12 |
| | ELOVL2 | chr6:10996933:C:T | rs9295741 | 0.15 | 0.02 | 3.28E-20 |
| osbond acid, all-cis-4,7,10,13,16-DPA;FFA_Docosapentaenoic Acid | FADS1-3 | chr11:61815236:T:C | rs174561 | −0.16 | 0.02 | 6.60E-19 |
| | SLCO1B1 | chr12:21178615:T:C | rs4149056 | 0.19 | 0.02 | 9.86E-16 |
| | ELOVL2-AS1 | chr6:11061917:C:T | rs9380073 | 0.13 | 0.02 | 2.86E-11 |
| | PNPLA3 | chr22:43928850:C:T | rs738408 | 0.13 | 0.02 | 4.21E-12 |

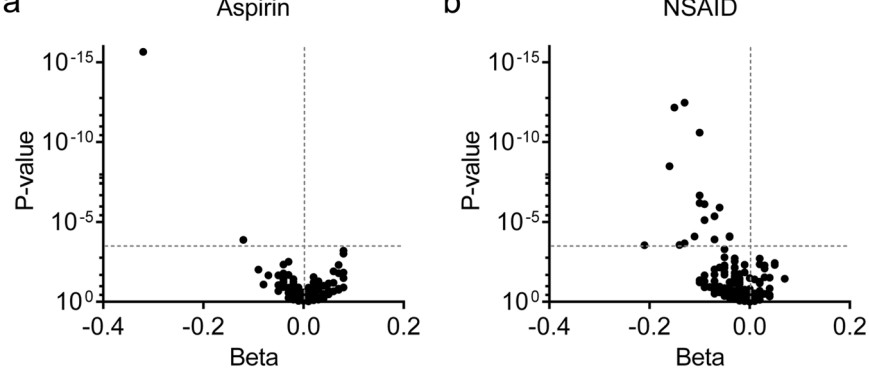

**Fig. 2 Eicosanoid associations with aspirin and NSAID use.** Volcano plots depicting the beta-coefficient (x-axis) and P-value (y-axis) of association with aspirin use (**a**) or NSAID use (**b**) for each eicosanoid in linear regression models adjusted for age, sex, race, study center, estimated glomerular filtration rate, antihypertension medications, systolic blood pressure, diabetes, smoking status, body mass index, atherosclerotic cardiovascular disease, cholesterol, HDL cholesterol and NSAID use (for aspirin analysis) or aspirin use (for NSAID analysis).

**Table 4 Eicosanoids associated with aspirin and NSAID use.**

| Eicosanoid | Beta aspirin | P-value aspirin | Beta NSAID | P-value NSAID | Locus | rsid | Beta SNP | P-value SNP |
|---|---|---|---|---|---|---|---|---|
| 20cooh AA_c | −0.04 | 4.02E-02 | −0.13 | 3.36E-13 | FADS1-3 | rs174564 | −0.38 | 1.02E-114 |
| | | | | | SLCO1B1 | rs11045885 | 0.16 | 1.64E-16 |
| | | | | | CYP4Z2P/CYP4A11 | rs6687264 | −0.30 | 1.46E-42 |
| 13S-HpOTrE(gamma) | −0.04 | 5.15E-02 | −0.15 | 6.82E-13 | UGT2B7 | rs28712409 | −0.12 | 9.58E-14 |
| | | | | | CYP4A11 | rs1126742 | −0.30 | 1.61E-49 |
| 13S-HpOTrE; 9S-HpOTrE | −0.04 | 4.59E-03 | −0.10 | 2.59E-11 | | | | |
| 9S-HpOTrE | −0.04 | 8.87E-02 | −0.16 | 3.27E-09 | SLCO1B1 | rs4149056 | 0.20 | 1.41E-17 |
| | | | | | LINC01835/CYP4F36P | rs77420750 | −0.15 | 9.96E-17 |
| 12S-HpETE_c | −0.04 | 1.67E-02 | −0.10 | 2.09E-07 | FADS1-3 | rs174557 | −0.12 | 4.66E-11 |
| 13-HpODE | −0.05 | 2.20E-02 | −0.10 | 6.38E-07 | FADS1-3 | rs174564 | 0.12 | 1.30E-12 |
| | | | | | SLCO1B1 | rs4149056 | 0.28 | 2.92E-34 |
| | | | | | CYP4B1/CYP4Z2P | rs4660960 | 0.14 | 9.00E-11 |
| | | | | | CYP4F2 | rs2108622 | −0.14 | 8.12E-15 |
| 5S-HpEPE | −0.04 | 2.76E-02 | −0.09 | 7.45E-07 | CYP4F2 | rs2108622 | −0.14 | 7.78E-14 |
| | | | | | FADS1-3 | rs28456 | 0.13 | 1.49E-13 |
| | | | | | SLCO1B1 | rs4149056 | 0.24 | 3.70E-24 |
| 12S-HpETE_a | −0.02 | 5.69E-02 | −0.06 | 1.21E-06 | | | | |
| 5S-HpETE_c | −0.01 | 7.19E-01 | −0.07 | 4.28E-06 | SLCO1B1 | rs4149056 | −0.21 | 8.41E-19 |
| 13-HpODE; 9-HpODE | 0.00 | 8.64E-01 | −0.09 | 7.53E-06 | | | | |
| 5,6-diHETrE | −0.01 | 1.27E-01 | −0.04 | 7.73E-05 | CYP2C8/ LOC107984257 | rs528961621 | 0.19 | 1.03E-16 |
| 8S-HETrE | −0.05 | 9.06E-02 | −0.11 | 8.02E-05 | | | | |
| 17-HETE; 18(+/-)-HETE; 20-HETE | 0.00 | 8.28E-01 | −0.04 | 8.42E-05 | CYP4Z2P/CYP4A11 | rs4507958 | −0.31 | 3.02E-52 |
| | | | | | TMEM258 | rs102274 | −0.18 | 5.75E-25 |
| | | | | | ADH1A/ADH1B | rs1693458 | −0.13 | 1.97E-10 |
| 17S-HpDHA | −0.02 | 2.30E-01 | −0.07 | 1.22E-04 | FADS1-3 | rs174583 | 0.11 | 9.68E-11 |
| 9-oxoOTrE | 0.01 | 7.04E-01 | −0.13 | 2.12E-04 | FADS1-3 | rs174564 | 0.13 | 2.89E-14 |
| 12-HHTrE | −0.32 | 2.22E-16 | −0.14 | 2.71E-04 | | | | |
| 10-nitroleate; 9-nitroleate | −0.12 | 1.31E-04 | 0.00 | 9.63E-01 | | | | |

expression of SCD[28]. In our study, we find that rs603424 is significantly associated with 11 eicosanoids, extending this potential cross-talk to include MUFAs and PUFA-derived eicosanoids—this is further reinforced by eicosanoid associations at several other genes involved in fatty acid and triglyceride metabolism, with the broader implication that changes in eicosanoid levels may contribute to the physiological and biological sequelae of altered nonessential fatty acid metabolism.

Because eicosanoids are most often viewed as acting acutely and in close proximity to their site of production, i.e., locally in tissue before leaking into the vasculature, how they ultimately undergo net excretion from circulation has garnered little attention. Our GWAS highlights various transport and conjugation mechanisms that likely participate in this latter process, with particular emphasis on hepatic excretion. In particular, variants in the liver-specific transporter encoded by SLCO1B1 are associated with numerous eicosanoids. Importantly, the index SNP in SLCO1B1, rs4149056, encodes a loss of function change, p.V174A. The transporter encoded by SLCO1B1 is best known for the transport of bilirubin, as corroborated by GWAS associations at this locus for serum bilirubin levels[29]. However, metabolomics GWAS have shown that this transporter is involved in the disposition of many compounds, including bile acids, lysophospholipids, and statin drugs[30, 31]. In vitro studies have suggested a role for this protein in prostaglandin transport as well[32], but this has not previously been demonstrated in humans. The p.V174A variant has known clinical importance, as carriers are at higher risk for statin-induced myopathy[31] and methotrexate-induced gastrointestinal toxicity[33].

Drugs that target eicosanoids are among the oldest and most commonly used in clinical medicine. Both aspirin and NSAIDs inhibit cyclooxygenases, suppressing the production of prostaglandins and thromboxanes. 12-HHTrE is a downstream product of the cyclooxygenase pathway, synthesized from PGH2 concurrently with TXA2, particularly in platelets[34]. Consistent with this, we find that aspirin use is associated with significantly lower levels of 12-HHTrE (with the association for NSAIDs at $P = 0.00027$ just missing the significance threshold). The biological role of 12-HHTrE is uncertain, with some studies highlighting a potential role in antagonizing TXA2 action[35]. We did not identify any significant associations with 12-HHTrE by GWAS, underscoring the strong environmental, i.e., pharmacologic, influence on its levels. We find that NSAID use is associated with reduced levels of fifteen eicosanoids, many of which also have significant genetic associations. For example, 20-carboxy arachidonic acid is downstream of cytochrome P450 metabolism of arachidonic acid, and its levels are associated with variants at CYP4Z2P/CYP4A11, as well as FADS1-3 and SCLO1B1, outlining potential mediators of pharmacogenomic variation.

Because the associations identified in our study recapitulate and potentially expand several key aspects of eicosanoid biochemistry, the absence of associations at loci encoding COX1, COX2, and lipoxygenases is noteworthy. It may be that some of these loci have a much greater impact on eicosanoids at a specific time and place, for example, the upregulation of COX2 expression in inflamed tissue, that is not captured by measurement of circulating levels in asymptomatic study participants. Alternatively, it is possible that deleterious variants at these loci have been subjected to negative selection or that common variants at these loci do not significantly affect downstream enzyme expression or function. Finally, it may be that some circulating eicosanoids are significantly associated with common variants at these loci but that they were not measured by our liquid chromatography-mass spectrometry method. However, several prostaglandins, thromboxanes, HpETEs, and leukotrienes

downstream of the encoded enzymes were assayed, including some that had other significant GWAS associations.

Several limitations of our study warrant mention. First, we do not have an independent replication cohort. However, we did conduct a meta-analysis across EA and AA study participants in the ARIC study and used a genome-wide significance threshold additionally adjusted for the number of examined analytes to attenuate the risk of false discovery. For many associations, strong biological plausibility further enhances confidence in the results. A second limitation is that the unambiguous classification and identification of some eicosanoids measured by our platform remains a challenge. We note that we have previously validated many of the measurements using a number of methodologies, including extensive chemical networking of mass spectral fragmentation and manual annotation of a subset of compounds[14]. With our nomenclature system, we acknowledge where analyte measurements may correspond to more than one isomer. Finally, it is known that eicosanoids are susceptible to non-enzymatic oxidation. Blood samples for this study were promptly stored at −80 °C following a standardized protocol, and our quality control analyses have demonstrated minimal artefactual contributions attributable to sample collection, storage, and processing[14]. We note that for any particularly sensitive analytes, such artefactual changes would bias association results toward null.

In summary, we identify 41 loci associated with 92 eicosanoids and related metabolites in >8000 ARIC Study participants, spanning both known and unknown determinants of eicosanoid metabolism. Future efforts will seek to replicate findings in independent cohorts, continue efforts to unambiguously annotate all measured eicosanoids, and probe the potential causal role of select locus-eicosanoid associations in disease, i.e., using Mendelian randomization as well as biological experimentation.

## Methods

### ARIC study.
The Atherosclerosis Risk in Communities (ARIC) study is a community-based prospective cohort study. Study participants were enrolled from Forsyth County, North Carolina, Jackson, Mississippi, suburbs of Minneapolis, Minnesota, and Washington County, Maryland, from 1987 to 1989. Blood eicosanoids were measured at a subsequent visit that occurred from 1990 to 1992 (visit 2). Participants who attended this visit, had blood eicosanoids measured and were free from end-stage kidney disease were included in the current study ($n = 9650$). All participants provided written informed consent, and the study adhered to the Declaration of Helsinki and was approved by the institutional review board of the Johns Hopkins University School of Medicine.

### Covariate definitions.
Covariates, including age, sex, self-reported race, study center, body mass index, systolic blood pressure, use of anti-hypertensive medications within the prior 2 weeks, estimated glomerular filtration rate, smoking status, and diabetes status were ascertained at the same study visit as blood eicosanoid measurements (visit 2). Systolic blood pressure was determined using three measurements with a random-zero sphygmomanometer, averaging the second and third measurements. Estimated glomerular filtration rate was estimated using the CKD Epidemiology Collaboration 2021 equation that includes both serum creatinine and cystatin C; creatinine was measured using the modified kinetic Jaffe method, and cystatin C was measured using a Roche Cobas 6000 chemistry analyzer. Smoking was self-reported. Antihypertension medications, aspirin use, and NSAID use were assessed as self-reported use within the 2 weeks prior to the study visit.

### Eicosanoid profiling.
Plasma samples for eicosanoid profiling were collected at visit 2 and immediately stored at −80 °C. Blood eicosanoids were measured using liquid chromatography-mass spectrometry, as previously described in detail[14, 36]. In brief, after undergoing both organic and solid phase extraction, samples were separated on a Phenomenex Kinetex C18 (1.7 μm, 100 × 2.1 cm) column using mobile phases A (70% water, 30% acetonitrile, 0.1% acetic acid) and B (50% acetonitrile, 50% isopropanol, 0.02% acetic acid) with a gradient starting at 1% B to 99% B over 8 min. Mass detection was performed using a Thermo QExactive orbitrap mass spectrometer in the negative ion mode. Data was collected using an MS1 scan event (scan range of m/z 225–650) followed by 4 DDA scan events using an isolation window of 1.0 m/z and a normalized collision energy of 35 arbitrary units. Quality control was evaluated by adjusting for technical variation in pooled plasma samples and internal standards spiked into each experimental sample as

well as assessing the coefficient of variation across 392 blind duplicate pairs. Missing eicosanoid levels were imputed with half of the minimum value for each individual eicosanoid, and eicosanoids missing in more than 50% of the samples were dropped, leaving 223 eicosanoids for statistical analysis. Eicosanoid levels were $\log_2$ transformed because of skewed distributions, and values outside of 5SDs from the mean were Winsorized. The median number of outliers was three per eicosanoid, ranging from 0 to 132.

### GWAS.
Genome-wide association studies were separately performed in EA and AA participants and combined using fixed-effect meta-analysis. Genotyping was performed using the Affymetrix 6.0 DNA microarray. Single-nucleotide polymorphisms (SNPs) with call rates <95%, Hardy–Weinberg equilibrium $P < 0.001$, or minor allele frequencies <1% were excluded[37]. The Trans-Omics for Precision Medicine reference (Freeze 5b) was used for data imputation[38–40]. There were 6496 white participants and 1910 black participants with both eicosanoid and genotype data. Eicosanoids were log-transformed and regressed on age, sex, and the first ten genetic and eicosanoid principal components. In a sensitivity analysis for significant associations, eicosanoids were log-transformed and regressed on age, sex, the first ten genetic and eicosanoid principal components, and aspirin and NSAID use. Residuals of these regressions were inverse-rank normalized and used as the dependent variable in GWAS using Fast Association Tests software. The statistical significance was set at a threshold of $5 \times 10^{-8}/223$ ($2.24 \times 10^{-10}$) according to the Bonferroni adjustment. For each eicosanoid, we identified the index SNP as the variant with the lowest $P$-value within a 1 Mb genomic radius. Index SNPs were annotated through linkage with the SNiPA web tool based on the 1000 Genomes phase 3 v5 and Ensembl v87 datasets. A genetic relationship matrix was calculated from all autosomal SNPs with an imputation quality of r2 > 0.6 using GCTA-GRM[71]. GCTA-GREML[72] was then used to estimate the proportion of variation in log2-transformed eicosanoid levels that can be explained by the SNPs for all eicosanoids.

### Transcriptome-wide association studies.
To provide additional support for the annotation of identified GWAS loci and to suggest potential tissue-specific sites of action, we performed transcriptome-wide association studies (TWAS) using models from the GTEx project v8 (http://gusevlab.org/projects/fusion/)[41]. We used Bonferroni correction to determine statistical significance, accounting for the number of investigated genes and tissues across the loci. We allowed the model to determine the best fit for each locus using the eicosanoid GWAS summary statistics.

### Statistics and reproducibility.
Baseline characteristics were summarized using mean, standard deviation, or median, 25th and 75th percentile, as indicated. Binary variables were summarized using percentages. We examined cross-sectional associations of eicosanoids with self-reported regular non-steroidal anti-inflammatory drugs and aspirin use in the 2 weeks preceding the study visit using linear regression. Associations of $\log_2$-transformed eicosanoids with medication use were examined using a model adjusted for age, sex, race and study center, smoking status, cholesterol, HDL cholesterol, diabetes, systolic blood pressure, antihypertension medication, body mass index, atherosclerotic cardiovascular disease, NSAIDs or aspirin, and estimated glomerular filtration rate. We used Bonferroni correction to account for multiple testing, dividing 0.05 by the number of eicosanoids investigated.

### Reporting summary.
Further information on research design is available in the Nature Portfolio Reporting Summary linked to this article.

## Data availability
Summary level data have been submitted to the NHGRI-EBI catalog (https://www.ebi.ac.uk/gwas/home) under GCP000680. The informed consent given by ARIC study participants does not cover data posting in public databases. However, data are available upon request from ARIC (https://sites.cscc.unc.edu/aric/contact_the_coord_center). Data requests can be submitted online and are subject to approval by the ARIC Steering Board.

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

## Acknowledgements

The work of P.S. was supported by the German Research Foundation (DFG) Project-ID 431984000 (CRC 1453), 192904750 (CRC 992), and 523737608 (SCHL 2292/2-1). The Atherosclerosis Risk in Communities study has been funded in whole or in part with federal funds from the National Heart, Lung, and Blood Institute, National Institutes of Health, Department of Health and Human Services, under Contract nos. (HHSN268201700001I, HHSN268201700002I, HHSN268201700003I, HHSN268201700005I, HHSN268201700004I). The authors thank the staff and participants of the Atherosclerosis Risk in Communities study for their important contributions.

## Author contributions

E.P.R., A.L.S., and M.E.G. designed the study. A.L.S. and P.S. performed the statistical analysis. E.P.R., A.L.S., P.S., and J.C. analyzed the data. M.A., M.J., and S.C. generated the eicosanoid data. Y.Y., J.C., B.Y., and M.E.G. contributed genetic data and analysis. E.P.R., A.L.S., and M.E.G. wrote the manuscript with input from all authors.

## Competing interests

M.J. holds significant interest and position at Sapient Bioanalytics, LLC, for work unrelated to the current manuscript. All remaining authors declare no competing interests.
