## [Peer Review File · Communications Biology]

Reviewers' comments:

Reviewer #1 (Remarks to the Author):

Rhee and colleagues have provided a manuscript entitled "Mechanisms of eicosanoid metabolism identified through genome-wide association study". The authors have performed a genome-wide association study in 8,406 participants of the ARIC Study. The manuscript is well written (I wish I could stress this more; it was very clear and easy to read). The results are well described, incorporating a discussion of biology not typically seen in metabolite GWAS.

Eicosanoids from plasma samples were extracted and quantified by high resolution LC-MS. The methodology has been published previously and used in several studies. I do thank the reviewers for including a section on eicosanoid profiling (something that is often missing in metabolite GWAS studies). The sample pre-processing is described and appropriate – including imputation with half the minimum value and winsorization.

Overall, I would recommend this manuscript for publication. Having said that, I have included several comments for the authors to consider.

The GWAS method is adequately described and appropriate. However, I do note that the authors say they adjust for the eicosanoid principal components (I assume the first 10). My concern is the authors may be regressing out a considerable amount of biology doing this and therefore losing power to detect associations. The reason for including the major genomic PCs is to try to capture artifacts that could induce false positive associations (such as population structure). As shown previously (<https://www.ncbi.nlm.nih.gov/pmc/articles/PMC5608948/>), metabolomic data generally does not contain such artefacts. The first metabolomic PCs will generally contain biology and systematic unwanted variation (only one of which you generally want to remove). Similarly, due to the strong correlation between metabolites, the first 10 PCs could represent a substantial proportion of variation for the whole dataset.

It would have been good to have seen heritability estimates included in the manuscript. As the authors have access to raw eicosanoid and genotype data, a GREML (or similar analysis) would be best. The Manhattan plots in the supp figure appear to mostly show mono- or oligo-genic associations, which may bias estimates using LD-score regression.

Minor comments:

In the introduction (lines 62-64), the authors state that no GWAS has been performed on blood eicosanoids. This isn't quite true. Back in 2020, a GWAS preprint was published (<https://www.medrxiv.org/content/10.1101/2020.08.01.20166413v1.full-text>) that consisted of many eicosanoids and oxylipin species. Similarly, the Metabolon HD4 does routinely measure several eicosanoids, which have appeared in many published GWAS.

I appreciate including the Manhattan plots as supp figures. However, I noticed a few issues with viewing these in excel (it could entirely be an issue with my computer). It would be great to see these available in PDF format when the manuscript is published.

The authors note that aspirin/NSAID medication is associated with eicosanoids. Have the authors investigated whether this should be accounted for in the GWAS? I know it isn't simple to adjust for medication usage in GWAS, but it would be interesting to know if this may have contributed to a decrease in power.

In the section on line 286, it is also possible that deleterious variants in COX1 and COX2 have been subject to selection. The authors could investigate this using GNOMAD, which may provide some additional insight.

Reviewer #2 (Remarks to the Author):

This study performed genome-wide association study of 223 eicosanoids in the ARIC dataset. It has the strength of covering both European and African ancestries. The study applied standard statistical approaches, with proper consideration of multiple testing correction. The results are clearly organized

and discussed in a biologically informative way, providing mechanistic insights into eicosanoid metabolism. It is pleasant to read this manuscript. The findings make significant contribution to our current understanding of the genetic basis of circulating eicosanoids. I am supportive of publishing this study after the following concerns are addressed.

- The lack of a replication dataset is a weakness. However, this dataset has a natural replication setup, the European and African ancestries. It is a huge missed opportunity to evaluate the reproducibility of genetic associations and also to evaluate the possibility of ancestry-specific effects. Moreover, the current manuscript only presented meta-analysis results, but had no mention of any results from either ancestry. It will be helpful to present the ancestry-specific association results for all significant loci in meta-analysis. That is, the beta and SE from both ancestries could be added to Supplementary Table 2. Please provide basic description of results from either ancestry, such as how many loci are significant only in Europeans or in Africans. Furthermore, there are multiple ways to evaluate the reproducibility. First, it will be informative to evaluate how many of the significant loci in Europeans are also significant in Africans (replication p value = 0.05 / # of loci). Second, it will be informative to compare the effect sizes across the two ancestries.
- Line 281 – 283, “For eicosanoids with significant genetic associations, including 20-carboxy arachidonic acid, genetic variants have as strong if not stronger influence on blood levels than NSAID use”. It is unclear how did the author make the comparison of genetic effect size to the NSAID effect size. Notably, the phenotypic transformations were different in the GWAS (log₂ transformed, regressed on covariates, then inverse-rank normalized) and the medication association analysis (only log₂ transformed). The effect sizes are not directly comparable between the two analyses.
- Line 367 – 368, “Eicosanoids were log-transformed and regressed on age, sex, and the first ten genetic and eicosanoid principal components (PCs).” It is unclear what “eicosanoid PCs” mean. If the eicosanoid PCs were derived from all eicosanoids, some of them would be highly correlated with the eicosanoid in a specific test and reduce the power to detect genetic effects. Please clarify this and provide the support to add these 10 eicosanoid PCs. Can you pick one metabolite and compare the results with or without adding these 10 eicosanoid PCs.
- Data availability, “The datasets generated during and/or analyzed during the current study are available from the corresponding author on reasonable request.” This statement is unacceptable in an era of transparency and reproducibility. First, please provide a statement about how to get access to the individual-level data, like dbGaP. Second, please release all GWAS summary statistics, as a common practice nowadays.

Minor comments

- Line 40, “medication-association eicosanoids” should be “medication-associated eicosanoids”.

Reviewer #1

Rhee and colleagues have provided a manuscript entitled “Mechanisms of eicosanoid metabolism identified through genome-wide association study”. The authors have performed a genome-wide association study in 8,406 participants of the ARIC Study. The manuscript is well written (I wish I could stress this more; it was very clear and easy to read). The results are well described, incorporating a discussion of biology not typically seen in metabolite GWAS. Eicosanoids from plasma samples were extracted and quantified by high resolution LC-MS. The methodology has been published previously and used in several studies. I do thank the reviewers for including a section on eicosanoid profiling (something that is often missing in metabolite GWAS studies). The sample pre-processing is described and appropriate – including imputation with half the minimum value and winsorization. Overall, I would recommend this manuscript for publication. Having said that, I have included several comments for the authors to consider.

Thank you for these encouraging comments.

The GWAS method is adequately described and appropriate. However, I do note that the authors say they adjust for the eicosanoid principal components (I assume the first 10). My concern is the authors may be regressing out a considerable amount of biology doing this and therefore losing power to detect associations. The reason for including the major genomic PCs is to try to capture artifacts that could induce false positive associations (such as population structure). As shown previously (<https://www.ncbi.nlm.nih.gov/pmc/articles/PMC5608948/>), metabolomic data generally does not contain such artefacts. The first metabolomic PCs will generally contain biology and systematic unwanted variation (only one of which you generally want to remove). Similarly, due to the strong correlation between metabolites, the first 10 PCs could represent a substantial proportion of variation for the whole dataset.

Thank you for this comment. Yes, we adjusted for the first 10 eicosanoid PCs. In our prior metabolomics GWAS [Kidney Int 2022;101(4):814-823], we have found that adjusting for metabolite PCs has been helpful for reducing unwanted systematic variation attributable to batch effects and other analytical factors. However, we appreciate the Reviewer’s perspective that this adjustment may also reduce power for genetic discovery. To test this directly, we repeated our GWAS adjusted for age, sex, and the first 10 genetic PCs, but NOT the first 10 eicosanoid PCs: with this approach, we identified 121 significant SNP-eicosanoid associations, compared to the 139 identified using the full adjustment presented in our manuscript, with substantial overlap. Notably, several of the significant associations absent in the GWAS not adjusted for eicosanoid PCs are either known positive associations, e.g. the association at *ELOVL* with FFA_Eicosapentaenoic Acid_c, FFA_Docosahexaenoic acid, and FFA_Adrenic Acid_a, or at loci with highly biologically plausible roles in eicosanoid metabolism, including *CYP2C9*, *CYP3A5*, and *CYP2C18*. Therefore, we believe that on balance, our adjustment for eicosanoid PCs is attenuating unwanted systematic variation more than it is attenuating true biologic signals, and have elected to show results for the fully adjusted model in our manuscript.

It would have been good to have seen heritability estimates included in the manuscript. As the authors have access to raw eicosanoid and genotype data, a GREML (or similar analysis) would be best. The Manhattan plots in the supp figure appear to mostly show mono- or oligo-genic associations, which may bias estimates using LD-score regression.

As suggested, we have now calculated heritability using GREML; this analysis was done in EA study participants because of the larger sample size. Heritability estimates are now provided for all eicosanoids in Supplementary Table 2 (all significant hits)—for these eicosanoids, mean heritability is 8.75% and median heritability is 7.31%.

Minor comments:

In the introduction (lines 62-64), the authors state that no GWAS has been performed on blood eicosanoids. This isn't quite true. Back in 2020, a GWAS preprint was published (<https://www.medrxiv.org/content/10.1101/2020.08.01.20166413v1.full-text>) that consisted of many eicosanoids and oxylipin species. Similarly, the Metabolon HD4 does routinely measure several eicosanoids, which have appeared in many published GWAS.

Thank you. We have downplayed our statement of novelty, instead stating: "By contrast, GWAS of blood metabolites have been more limited..."

I appreciate including the Manhattan plots as supp figures. However, I noticed a few issues with viewing these in excel (it could entirely be an issue with my computer). It would be great to see these available in PDF format when the manuscript is published.

We are happy to provide these plots in PDF format, of course deferring to the publisher's best practices.

The authors note that aspirin/NSAID medication is associated with eicosanoids. Have the authors investigated whether this should be accounted for in the GWAS? I know it isn't simple to adjust for medication usage in GWAS, but it would be interesting to know if this may have contributed to a decrease in power.

This is an interesting question. In a sensitivity analysis, we repeated our GWAS using models further adjusted for aspirin and NSAID use. The total number of significant associations was very similar (136 with medication adjustment versus 139 without medication adjustment). Further, for all significant associations in our manuscript, the effect size of association was essentially unchanged in the GWAS models adjusted for medication use—this is shown in the Figure below, which is now included as Supplementary Figure 3 in the revised manuscript.

Supplementary Figure 3. Comparison of GWAS with or without medication adjustment.

Scatter plot of effect sizes of significant loci in main GWAS analysis (x-axis) versus effect sizes in GWAS further adjusted for aspirin and NSAID use (y-axis).

In the section on line 286, it is also possible that deleterious variants in COX1 and COX2 have been subject to selection. The authors could investigate this using GNOMAD, which may provide some additional insight.

Thank you for this comment. We have edited the pertinent text in the Discussion to state, "Alternatively, it is possible that deleterious variants at these loci have been subjected to negative selection or that common variants at these loci do not significantly affect downstream enzyme expression or function."

Reviewer #2

This study performed genome-wide association study of 223 eicosanoids in the ARIC dataset. It has the strength of covering both European and African ancestries. The study applied standard statistical approaches, with proper consideration of multiple testing correction. The results are clearly organized and discussed in a biologically informative way, providing mechanistic insights into eicosanoid metabolism. It is pleasant to read this manuscript. The findings make significant contribution to our current understanding of the genetic basis of circulating eicosanoids. I am supportive of publishing this study after the following concerns are addressed.

Thank you for these encouraging comments.

The lack of a replication dataset is a weakness. However, this dataset has a natural replication setup, the European and African ancestries. It is a huge missed opportunity to evaluate the reproducibility of genetic associations and also to evaluate the possibility of ancestry-specific effects. Moreover, the current manuscript only presented meta-analysis results, but had no mention of any results from either ancestry. It will be helpful to present the ancestry-specific association results for all significant loci in meta-analysis. That is, the beta and SE from both ancestries could be added to Supplementary Table 2. Please provide basic description of results from either ancestry, such as how many loci are significant only in Europeans or in Africans. Furthermore, there are multiple ways to evaluate the reproducibility. First, it will be informative to evaluate how many of the significant loci in Europeans are also significant in Africans (replication p value = 0.05 / # of loci). Second, it will be informative to compare the effect sizes across the two ancestries.

Thank you for this comment. As suggested by the Reviewer, we now include ancestry-specific information on beta, SE, and P-value for all significant associations identified by meta-analysis in the revised Supplementary Table 2. In the Results, we note: "When the EA and AA cohorts were analyzed separately, 9 of 41 loci identified by meta-analysis had significant associations in both the EA and AA cohorts, 25 had significant associations in the EA cohort only, 1 had a significant association in the AA cohort only, and 6 did not have an association in either the EA or AA cohort (Supplementary Table 2)."

We believe the greater number of significant associations in the EA cohort is likely due to increased power (6496 self-identified EA study participants and 1910 self-identified AA study participants). Importantly, the effect sizes (Betas) were highly consistent for all of the associations shown in Supplementary Table 2 across ancestry (new Supplementary Figure 1).

Supplementary Figure 1. Comparison of effect sizes at significant loci by ancestry. Scatter plot of effect sizes of significant eicosanoid GWAS loci in EA (x-axis) versus AA (y-axis) cohorts.

Line 281 – 283, “For eicosanoids with significant genetic associations, including 20-carboxy arachidonic acid, genetic variants have as strong if not stronger influence on blood levels than NSAID use”. It is unclear how did the author make the comparison of genetic effect size to the NSAID effect size. Notably, the phenotypic transformations were different in the GWAS (log₂ transformed, regressed on covariates, then inverse-rank normalized) and the medication association analysis (only log₂ transformed). The effect sizes are not directly comparable between the two analyses.

Thank you for this astute comment. We made this statement based on the stronger P-values of association for select SNPs than for medications, but we agree with the Reviewer that these are not directly comparable. Therefore, we have removed this statement.

Line 367 – 368, “Eicosanoids were log-transformed and regressed on age, sex, and the first ten genetic and eicosanoid principal components (PCs).” It is unclear what “eicosanoid PCs” mean. If the eicosanoid PCs were derived from all eicosanoids, some of them would be highly correlated with the eicosanoid in a specific test and reduce the power to detect genetic effects. Please clarify this and provide the support to add these 10 eicosanoid PCs. Can you pick one metabolite and compare the results with or without adding these 10 eicosanoid PCs.

Please note we provide a very similar response to Reviewer #1’s first comment. Yes, eicosanoid PCs were derived from all eicosanoids. In our prior metabolomics GWAS [Kidney Int 2022;101(4):814-823], we have found that adjusting for metabolite PCs in this way has been helpful for reducing unwanted systematic variation attributable to batch effects and other analytical factors. However, we appreciate both Reviewers’ perspectives that this adjustment may also reduce power for genetic discovery. To test this directly, we repeated our GWAS adjusted for age, sex, and the first 10 genetic PCs, but NOT the first 10 eicosanoid PCs: with this approach, we identified 121 significant SNP-eicosanoid associations, compared to the 139 identified using the full adjustment presented in our manuscript, with substantial overlap. Notably, several of the significant associations absent in the GWAS not adjusted for eicosanoid PCs are either known positive associations, e.g. the association at *ELOVL* with FFA_Eicosapentaenoic Acid_c, FFA_Docosahexaenoic acid, and FFA_Adrenic Acid_a, or at loci with highly biologically plausible roles in eicosanoid metabolism, including *CYP2C9*, *CYP3A5*,

and *CYP2C18*. In addition, we show results for the top 4 associations identified in our paper with or without adjustment for eicosanoid PCs in the Table below—as shown, removing the adjustment for eicosanoid PCs does not systematically strengthen associations.

Locus	Eicosanoid	rsid	Age, sex, genetic and eicosanoid PC		Age, sex, genetic PCs	
			Beta	P-value	Beta	P-value
FADS1-3	FFA_Arachidonic Acid_a	rs174544	0.54	6.60E-207	0.27	5.28E-49
SLCO1B1	11t LTD4	rs4149056	0.54	1.47E-124	0.70	1.95E-219
PKD2L1	11-hydroxy-9-octadecenoate; 10-hydroxy-11-octadecenoate	rs603424	-0.41	1.94E-113	-0.39	6.21E-100
ACOT4/ACOT6	Dihydroxydocosapentaenoic acid	rs111511359	-0.34	1.99E-64	-0.34	5.63E-62

Therefore, we believe that on balance, our adjustment for eicosanoid PCs is attenuating unwanted systematic variation more than it is attenuating true biologic signals, and have elected to show results for the fully adjusted model in our manuscript.

Data availability, “The datasets generated during and/or analyzed during the current study are available from the corresponding author on reasonable request.” This statement is unacceptable in an era of transparency and reproducibility. First, please provide a statement about how to get access to the individual-level data, like dbGaP. Second, please release all GWAS summary statistics, as a common practice nowadays.

Thank you for this important feedback. The Data Availability statement has been changed to: “Summary level data will be made available at the NHGRI-EBI catalog (<https://www.ebi.ac.uk/gwas/home>). The informed consents given by ARIC study participants do not cover data posting in public databases. However, data are available upon request from ARIC (https://sites.csc.unc.edu/aric/contact_the_coord_center). Data requests can be submitted online and are subject to approval by the ARIC Steering Board.”

Minor comments

Line 40, “medication-association eicosanoids” should be “medication-associated eicosanoids”.

This change has been made.

REVIEWERS' COMMENTS:

Reviewer #1 (Remarks to the Author):

The authors have addressed all my comments. Furthermore, they have included additional analyses in the supplementary section, which improves the manuscript.
I recommend this for publication.

Reviewer #2 (Remarks to the Author):

The authors did an outstanding job addressing my concerns. I support the publication of the manuscript in its current form.